# Multi-modal Cell Segmentation based on U-Net++ and Attention Gate

**Xinye Yang, Hao Chen, Lihua Huang and Xuru Zhang**
College of Physics and Information Engineering
Fuzhou University

**Liqin Huang**
College of Physics and Information Engineering
Fuzhou University
hlq@fzu.edu.cn

## Abstract

Cell segmentation is one of the most fundamental tasks in the areas of medical image analysis, which assists in cell recognition and number counting. The segmentation results obtained will be poor due to the diverse cell morphology and the frequent presence of impurities in the cell pictures. In order to solve the cell segmentation which are from a competition held by Neural Information Processing Systems(NIPS), we present a network that combines attention gates with U-Net++ to segment varied sizes of cells. Using the feature filtering of the attention gate can adjust the convolution block's output, so as to improve the segmentation effect. The F1 score of our method reached 0.5874, Rank Running Time get 2.5431 seconds.

## 1 Introduction

Cell segmentation refers to dividing the cell image into several disjoint regions according to the characteristics of gray, color, and geometry, so that these features show consistency or similarity in the same regions. Traditional cell segmentation methods used pixel-level processing methods such as morphology and gray value to segment cells. Threshold-based segmentation was one of the most commonly used methods, because it had efficient computational and stable performance [1]. However, it only considered the pixel, not to the spatial characteristics of the image, resulting in very sensitivity to noise. With the deepening of research, several new cell segmentation algorithms had emerged, such as domain-based methods and active contour model-based methods [2]. Based on the edge detection of melanoma (tumor cells) and lymphocytes (blood cells), D.Anorganingrum et al. used a combination method of median filtering and mathematical morphological operations such as dilation and erosion for segmentation [3]. R.Arulmurugan and H.Anandakumar introduced a region-based cell detection and segmentation method [4], namely histogram color contrast seed point selection (HCC-SPS), which could group similar color values, so as to solve the color contrast problem in visual signals and generate accurate required edge points. The region-based seed points can fine-tune the salient values, making the difference between salient points and background points more obvious. J.M.Sharif et al. introduced a method of red blood cell (RBC) segmentation [5], involving YCbCr color conversion, mask, morphological operator, and watershed algorithm. The combination of YCbCr color conversion and morphological operator produced segmented white blood cell (WBC) nuclei, which were used as a mask to remove WBC from the blood cell image. Nevertheless, due to the complexity of the cell image, the uneven illumination of the microscopic image, the gray level change of the object itself and other problems, there were still remained several challenges in the segmented images, mainly including cell adhesion, cell overlap, holes, and so on.

36th Conference on Neural Information Processing Systems (NeurIPS 2022).

The development of deep learning (DL) promoted widely application of neural networks in image segmentation. Yang L et al. proposed a weakly supervised method combining graph search (GS) and DL for biomedical image segmentation using box annotations [6]. Saleem S et al. proposed an improved DL method, using a pre-trained deep model to extract deep features from each blood smear image for accurate segmentation and classification of white blood cells [7]. Based on convolutional neural networks (CNNs), DL can achieve a good performance in image processing. Eschweiler D et al. proposed a 3D segmentation method, which combined the discrimination ability of CNN for preprocessing [8]. Akram S U et al. proposed a method based on CNN, which can be used for cell detection, segmentation, and tracking [9]. Hatipoglu N et al. used special DL algorithms (including CNN, stacked autoencoder and deep belief networks), and spatial relations, adding local space and contextual information [10].

With the appearance of U-Net [11], its powerful performance in the medical image had attracted extensive attention. Many medical image semantic segmentation tasks adopted U-Net as the baseline. As shown in Figure (a) 1, the backbone of U-Net network is an encoder-decoder structure, whose encoder uses convolution for feature extraction, and the decoder performs convolution and up sampling. The U-Net had evolved into many new network structures. For instance, Huang H et al. introduced a novel U-Net++ [12], which made use of all-around skip connections and deep supervision. It was particularly useful for targets with different scales. In addition to improving accuracy, the proposed U-Net++ could also reduce the number of network parameters, thereby improving computing efficiency. Zhou Y et al. proposed a new architecture called dimensional fusion U-Net (DU-Net) [13], which innovatively combines 2D and 3D convolutions in the coding stage for chronic stroke lesion segmentation. Nabil Ibtehaz et al. proposed a combined path using $3 \times 3$ convolutional blocks and $1 \times 1$ convolutional blocks instead of skip connection in U-Net for the purpose of multiple residual connections [14].

The previous segmentation methods were usually limited to one modality, and the generalization performance of the trained model could not be guaranteed. In the weakly supervised cell segmentation competition held by NIPS, after observing dataset used in the competition, we found that the cells were morphologically diverse, different imaging colorants led to four different modals of cell data, such as different sizes and colors, which greatly tested the generalization ability of the model.

This paper attempted to improve the U-Net++ model by using a method based on the combination with attention. We designed a new U-Net++ model adding the attention gate. We had tested our network in this competition to verify that the proposed method was indeed capable of performing the segmentation task. We used a fully supervised network that focuses on the generalization performance of the network, having good robustness. The F1 performance of our model had reached 0.5874.

## 2   Method

We used the U-Net++ network to process the fusion of different scales of features, which was expected to capture cells of various shapes. In additon the distribution of cells in space was also chaotic, using attention gates can better help the network to select regions of interest.

### 2.1   U-Net++

U-Net had the following two major defects: (1) The optimal depth of the network was unknown. Experiments showed that a deeper U-Net may lead to even worse results. Therefore, it was necessary to integrate networks of different depths and made choices through a large number of experiments. (2) Skip connection introduced unnecessary restrictions, and feature fusion was only performed at the same scale. U-Net++ proposed by Zongwei Zhou et al. connected encoder's and decoder's subnets through a series of nested and dense skip paths, aiming to narrow the semantic gap between the feature maps of subnets [15]. The U-Net++ structure was shown in Figure 1 (b). Compared with U-Net, U-Net++ had embedded U-Net with different depths, so it had more flexible skip connections. U-Net++ can be seen as a network built by combining different layers of U-Net networks. Because different layers of U-Net paid different attention to feature extraction of the input image, U-Net++ can extract the features after various layers of U-Net, instead of only integrating for the same scale.

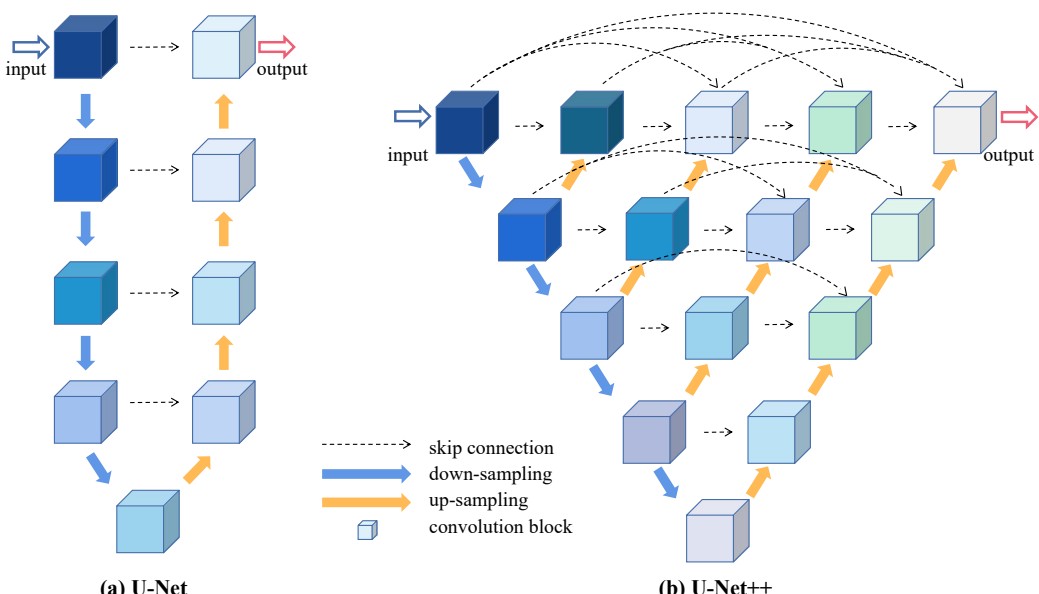

(a) U-Net          (b) U-Net++

Figure 1: There are the structure of 5-layer U-Net and 5-layer U-Net++. It can be seen that U-Net++ can be regarded as composed of various U-Net whose number of layers is less than 5.

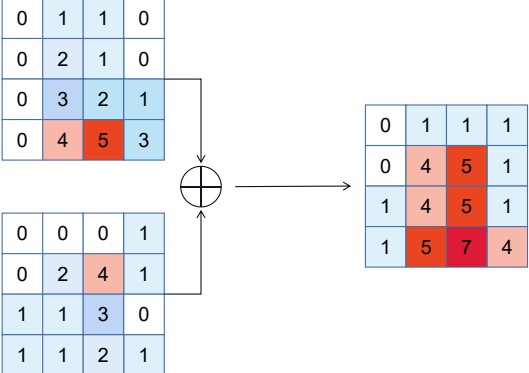

Figure 2: By adding different feature maps, the information of the same region of interest will be strengthened, and the different regions can also be used as auxiliary information. The two together will have more auxiliary information. This strategy emphasizes the core information without neglecting the details.

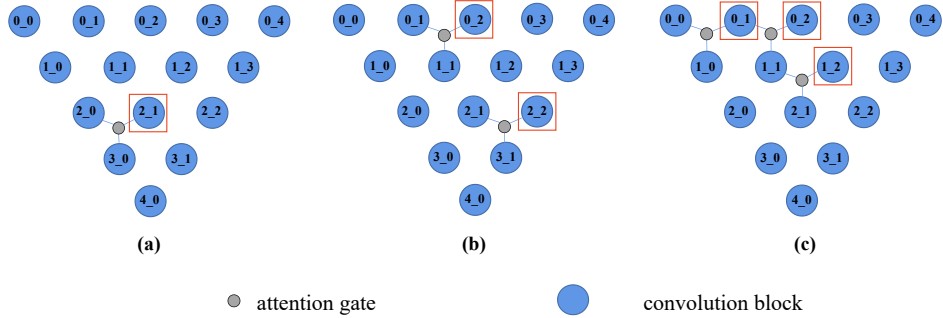

Figure 3: We use $G$ to uniformly represent attention gate. For each attention gate at specific location, its identification is consistent with the name of its output convolution block which marked by red box in this figure. In (a),we express the attention gate as $G^{2.1}$, then (b) is $G^{0.2} + G^{2.2}$, (c) is $G^{0.1}+G^{0.2}+G^{1.2}$.

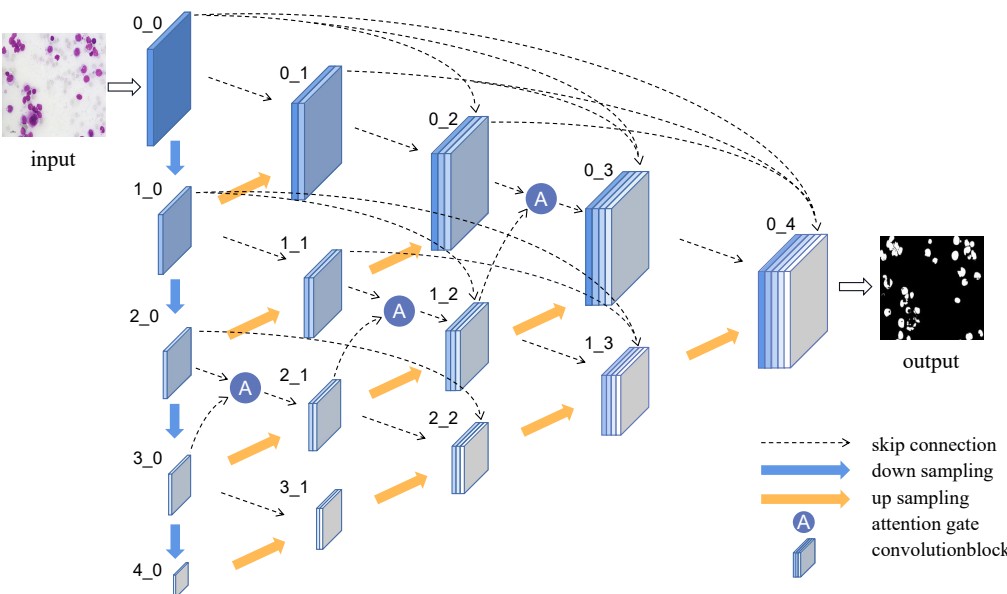

Figure 4: We follow the setting method of the previous attention gate. The input of the gate consists of two parts: one part is connected from the previous convolution block, and the other part is up sampling from the next convolution block.

## 2.2 Attention Gate

In recent years, with the development of attention mechanism [16], it had been widely used because of its excellent performance. Attention can capture the semantic relationship between elements in the sequence, and added weight to each of them so that subsequent training can capture key features. When it was applied to the 2D images, the relationship between the pixel and the position of the object in image can be obtained.

Generally, it is very difficult for CNN to predict the FP(false positive) of small targets, so the usual method is to locate them first and then segment them. CNN with attention gate[17] can also achieve this effect. Attention gate is used in natural image analysis, knowledge graphs, image description, machine translation and classification tasks. It was a plug-and-play model which can be added in front of the convolution block. It does not need to train multiple models and a lot of additional parameters. It can suppress the characteristic response in the unrelated background region, without the requirement to crop a ROI(region of interest) between networks. That was, before concatenating the features on each resolution of the encoder of U-Net++ with the corresponding features in the decoder, an attention gate was used to control the importance of features at different spatial locations, suppress the irrelevant areas in the input images, and highlight the features of specific local areas. The reason for adding and reprocessing the two inputs was that the extracted features of two feature map with the same size and number of channels after processing were different. The addition operation can strengthen the information of the same region of interest, and the different regions can also be used as auxiliary Figure 2. It was to further emphasize the core information without neglecting the details.

In general, two feature maps enter the attention gate model and calculate the attention weight matrix $A_t$, then take a element-wise multiplication between $A_t$ and one of the feature maps to get a final output. In the whole process, the output size and the input feature map is the same as $A_t$.

## 2.3 Proposed Method

In [17], attention gate was set in the U-Net decoding layer, while U-Net++ could be seen as a number of U-Net components. It meant that we could add attention gate in many places in U-Net++. So based on the U-Net++ framework, we added the attention module. We used $G$ to represent attention gate, the G position setting method and its expression were shown in Figure 3. Obviously, there were many methods of setting. After initial simple test, we adopted three attention gates in submitted method including $G^{1.2}$, $G^{2.1}$, $G^{0.3}$, as Figure 4. The three selected convolution kernels were respectively set as follows: $G^{2.1}$ was set as [128, 256, 256], $G^{1.2}$ was set to [64, 128, 128] and $G^{0.3}$ was [32, 64, 64]. Each attention gate accepted two convolution module inputs, which adapted to the fact that attention gate adopted two inputs for enhancing features.

## 2.4 Loss

We employed Dice Focal Loss, which was the weighted sum of Focal Loss and Dice loss. The expression of Focal Loss is as follows:

$$\mathcal{L}_{focal} = -\alpha \left(1 - p'_t\right)^\gamma \log p'_t \tag{1}$$

where $p'_t$ represents the probability of the predicted value, and the role of $\alpha$ is to weight the loss of different types of samples. If there are few positive samples, the weight of the loss of positive samples will be increased; The role of $\gamma$ is to determine the degree of attenuation. When the predicted value of the sample, that is, the $p_t$ of the easily distinguishable sample, is relatively large, the corresponding $1 - p_t$ will be very small, so that the loss of the easily distinguishable sample will be significantly reduced. Compared with the easily distinguishable sample, the decline of the hard distinguishable sample is less, and the model will pay more attention to the optimization of the loss of the hard distinguishable samples.

The expression of Dice loss is as follows:

$$\mathcal{L}_{Dice} = 1 - \frac{2 * |y \cap y'| + smooth_{up}}{|y| + |y'| + smooth_{down}} \tag{2}$$

where $|y|$ and $|y'|$ represent ground truth and predict mask respectively. $smooth_{up}$ and $smooth_{down}$ are modifiable items, this is to avoid the problem that when $|y|$ and $|y'|$ are both 0, the molecule is divided by 0 or the molecule is 0. It can also reduce over fitting.

The expression of Dice Focal Loss is as follows:

$$\mathcal{L} = \lambda_f * \mathcal{L}_{focal} + \lambda_d * \mathcal{L}_{Dice} \tag{3}$$

both $\lambda_d$ and $\lambda_f$ are weight value. We set them both 0.5.

## 2.5 Post Processing

We had adopted morphological processing method in the prediction module. Through experiments, the method of closing arithmetic, can make the prediction effect of the method used better. This was because the opening arithmetic may erode the small white spots predicted correctly and cannot be recovered. The specific experimental data were described in detail in the experiment section.

# 3 Experiments

The dataset were 1000 labeled image patches of various microscopy types, tissue types and staining types, and more than 1700 unlabeled images. There are four microscope modalities in the training set, including: brightfield (300 patches), fluorescent(300 patches), phase-difference (200 patches), and differential interference contrast (200 patches). There are 101 images for model testing. Due to the difference of tissues and staining methods, the style of images is varied. We conducted the following series of experiments according to the baseline provided on the official website. The development environments and requirements are presented in Table 1.

Table 1: Development environments and requirements.

| System | e.g., Ubuntu 20.04.4 LTS |
|---|---|
| CPU | Intel(R) Core(TM) i9-12900X |
| RAM | 62GB |
| GPU (number and type) | Two Tesla 13G |
| CUDA version | 11.7 |
| Programming language | Python 3.9 |
| Deep learning framework | Pytorch (Torch 1.8.1, torchvision 0.9.1) |

## 3.1 Data Augmentation

We used classical methods to preprocess the input data, such as random crop, random rotation, nonlinear transformation of intensity histogram.

## 3.2 Comparison with Some Other Methods

Some network parameter settings are shown in the Table 2. In Table 3,we compared our method to existing some other networks,i.e., U-Net [11], U-NetR [18], Swin U-NetR [19] and U-Net++ [15]. Effectively, the proposed methods achieved the best F1 score among all methods. Besides, the visual results of each network were shown in Figure 5. Particularly, one can see our proposed method achieved more robust results in the second row of Figure 5. Whereas, other methods were easily degraded by impurities, demonstrating the robustness of our method.

We used the teacher-student model to utilize the un-label images. The one with the best F1 score was selected as the teacher model from the all fully supervised models completed with training. The teacher model provided pseudo-label for the un-label images, which were used for the student model training. We selected the same networks used in the previous step as the student model for performance testing. The loss function we used was the consistency loss. The results were shown in Table 3, those marked with "(s)" are the student-models of the corresponding networks.

Table 2: Some network parameter

| | |
|---|---|
| Batch size | 8 |
| Patch size | 3×256×256 |
| Total epochs | 1000 |
| Optimizer | Adamw |
| Initial learning rate (lr) | 0.0001 |

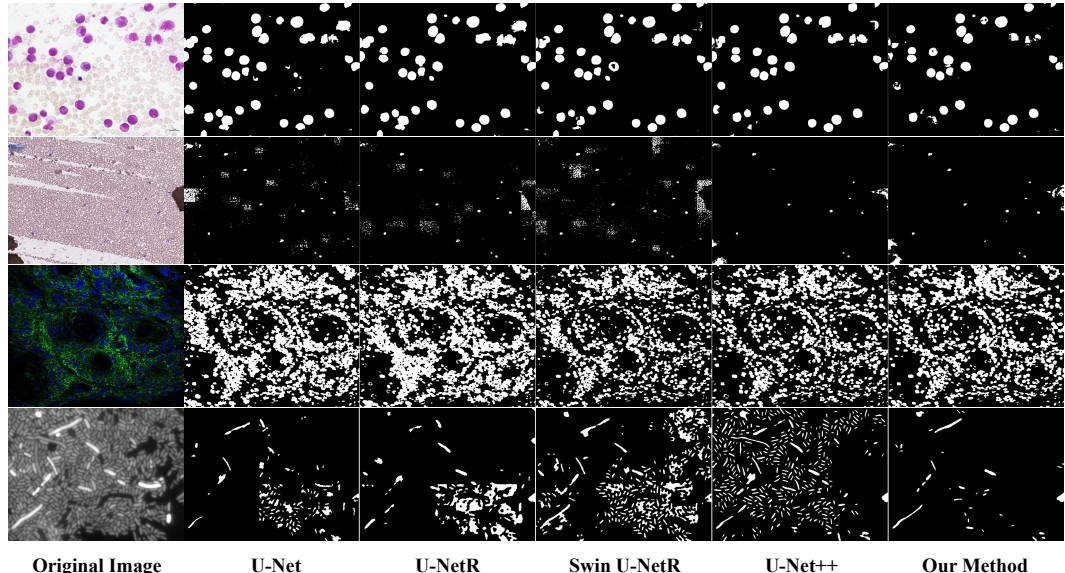

Figure 5: The visualization result of different cell segmentation networks on four types of cell images.

### 3.3 Study of Attention Gates with Different Quantities

In view of the impact of different numbers of attention gates, we conducted an ablation experiment. The position and corresponding naming of the attention gate can refer to Figure 3. The results was shown in Table 4. It can be seen from the table data that with more attention gates, F1 score didn't go up but down. This showed that when setting attention gates, increasing the quantities did not necessarily improve the performance. Increasing the quantities of attention gate may be counterproductive.

### 3.4 Post Processing

We tested the impact of opening and closing arithmetic on the prediction results. The results were submitted by the competition website for calculation. The F1 score obtained through the closing arithmetic was 0.5834, the open arithmetic is 0.5796, using both get 0.5597. Apparently, using closed arithmetic as post-processing can improve prediction performance.

### 3.5 Results on Final Testing Set

Table 5and Table 6 are our testing results in the NeurIPS cell segmentation challenge.

## 4 Conclusion and Discussion

This paper proposed a model combined with U-Net++ for the competition of weakly supervised cell segmentation in multi-modal high-resolution microscope images held by NIPS. Our method used the attention gate to increase the ROI processing of U-Net++ for input, thus improving the segmentation performance of the model for different modal data. The experimental results showed

Table 3: Performance of different cell segmentation networks

| Method | F1 Score |
|---|---|
| U-Net | 0.4879 |
| U-Net(s) | 0.2966 |
| U-NetR | 0.5174 |
| U-NetR(s) | 0.4136 |
| Swin U-NetR | 0.5588 |
| Swin U-NetR(s) | 0.4407 |
| U-Net++ | 0.5622 |
| U-Net++(s) | 0.4420 |
| Proposed method | **0.5874** |
| Proposed method(s) | 0.4467 |

Table 4: Performance of proposed network with different quantities of attention gates

| G | F1 Score |
|---|---|
| $G^{1.2}$ | 0.5691 |
| $G^{2.1}$ | 0.5767 |
| $G^{0.3}$ | 0.5624 |
| $G^{1.2}+G^{2.1}$ | 0.5593 |
| $G^{2.1}+G^{0.3}$ | **0.5972** |
| $G^{1.2}+G^{0.3}$ | 0.5738 |
| $G^{2.1}+G^{0.3}+G^{1.2}$ | 0.5874 |

that the proposed network method can further optimize cell segmentation on the basis of U-Net++. The F1 score our method tested reached 0.5874. Nevertheless, from the results, the small cell segmentation performance did not seem to have been improved. We will analyze and try to solve this situation in the future. Furthermore, we had tried to use the teacher-student model to train a semi-supervised segmentation model with un-label image, but the effect was not satisfactory. Then, we believed that there is a theoretical basis for how many and where to put the attention gate. However, we just did ablation experiments in the specific area. If someone wanted to test all the situations in which it can be placed in U-Net++, only for the U-Net++ we used in this experiment, there will be $2^{10} = 1024$ cases. It was a huge workload. How to find the most appropriate setting scheme was also a problem worth exploring.

# 5   Acknowledgement

The authors of this paper declare that the segmentation method they implemented for participation in the NeurIPS 2022 Cell Segmentation challenge has not used any private datasets other than those provided by the organizers and the official external datasets and pretrained models. The proposed solution is fully automatic without any manual intervention.

This work was supported by National Natural Science Foundation of China(62271149), Fujian Provincial Natural Science Foundation project(2021J02019).

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

Table 5: Testing Results of Median

| Median F1-All | Median F1-BF | Median F1-DIC | Median F1-Fluo | Median F1-PC |
|---|---|---|---|---|
| 0.2284 | 0.2883 | 0.4541 | 0.0158 | 0.3454 |

Table 6: Testing Results of Mean

| Mean F1-All | Mean F1-BF | Mean F1-DIC | Mean F1-Fluo | Mean F1-PC |
|---|---|---|---|---|
| 0.298 | 0.3071 | 0.4477 | 0.0721 | 0.44 |

[4] R Arulmurugan and H Anandakumar. Region-based seed point cell segmentation and detection for biomedical image analysis. International Journal of Biomedical Engineering and Technology, 27(4):273–289, 2018.

[5] Congcong Zhang, Xiaoyan Xiao, Xiaomei Li, Ying-Jie Chen, Wu Zhen, Jun Chang, Chengyun Zheng, and Zhi Liu. White blood cell segmentation by color-space-based k-means clustering. Sensors, 14(9):16128–16147, 2014.

[6] Lin Yang, Yizhe Zhang, Zhuo Zhao, Hao Zheng, Peixian Liang, Michael TC Ying, Anil T Ahuja, and Danny Z Chen. Boxnet: Deep learning based biomedical image segmentation using boxes only annotation. arXiv preprint arXiv:1806.00593, 2018.

[7] Saba Saleem, Javeria Amin, Muhammad Sharif, Muhammad Almas Anjum, Muhammad Iqbal, and Shui-Hua Wang. A deep network designed for segmentation and classification of leukemia using fusion of the transfer learning models. Complex & Intelligent Systems, 8(4):3105–3120, 2022.

[8] Dennis Eschweiler, Thiago V Spina, Rohan C Choudhury, Elliot Meyerowitz, Alexandre Cunha, and Johannes Stegmaier. Cnn-based preprocessing to optimize watershed-based cell segmentation in 3d confocal microscopy images. In 2019 IEEE 16th International Symposium on Biomedical Imaging (ISBI 2019), pages 223–227. IEEE, 2019.

[9] Saad Ullah Akram, Juho Kannala, Lauri Eklund, and Janne Heikkilä. Cell segmentation proposal network for microscopy image analysis. In Deep Learning and Data Labeling for Medical Applications, pages 21–29. Springer, 2016.

[10] Nuh Hatipoglu and Gokhan Bilgin. Cell segmentation in histopathological images with deep learning algorithms by utilizing spatial relationships. Medical & biological engineering & computing, 55(10):1829–1848, 2017.

[11] Olaf Ronneberger, Philipp Fischer, and Thomas Brox. U-net: Convolutional networks for biomedical image segmentation. In International Conference on Medical image computing and computer-assisted intervention, pages 234–241. Springer, 2015.

[12] Huimin Huang, Lanfen Lin, Ruofeng Tong, Hongjie Hu, Qiaowei Zhang, Yutaro Iwamoto, Xianhua Han, Yen-Wei Chen, and Jian Wu. Unet 3+: A full-scale connected unet for medical image segmentation. In ICASSP 2020-2020 IEEE International Conference on Acoustics, Speech and Signal Processing (ICASSP), pages 1055–1059. IEEE, 2020.

[13] Yongjin Zhou, Weijian Huang, Pei Dong, Yong Xia, and Shanshan Wang. D-unet: a dimension-fusion u shape network for chronic stroke lesion segmentation. IEEE/ACM transactions on computational biology and bioinformatics, 18(3):940–950, 2019.

[14] Nabil Ibtehaz and M Sohel Rahman. Multiresunet: Rethinking the u-net architecture for multimodal biomedical image segmentation. Neural networks, 121:74–87, 2020.

[15] Zongwei Zhou, Md Mahfuzur Rahman Siddiquee, Nima Tajbakhsh, and Jianming Liang. Unet++: A nested u-net architecture for medical image segmentation. In Deep learning in medical image analysis and multimodal learning for clinical decision support, pages 3–11. Springer, 2018.

[16] Ashish Vaswani, Noam Shazeer, Niki Parmar, Jakob Uszkoreit, Llion Jones, Aidan N Gomez, Łukasz Kaiser, and Illia Polosukhin. Attention is all you need. Advances in neural information processing systems, 30, 2017.

[17] Ozan Oktay, Jo Schlemper, Loic Le Folgoc, Matthew Lee, Mattias Heinrich, Kazunari Misawa, Kensaku Mori, Steven McDonagh, Nils Y Hammerla, Bernhard Kainz, et al. Attention u-net: Learning where to look for the pancreas. arXiv preprint arXiv:1804.03999, 2018.

[18] Ali Hatamizadeh, Yucheng Tang, Vishwesh Nath, Dong Yang, Andriy Myronenko, Bennett Landman, Holger R Roth, and Daguang Xu. Unetr: Transformers for 3d medical image segmentation. In Proceedings of the IEEE/CVF Winter Conference on Applications of Computer Vision, pages 574–584, 2022.

[19] Ali Hatamizadeh, Vishwesh Nath, Yucheng Tang, Dong Yang, Holger R Roth, and Daguang Xu. Swin unetr: Swin transformers for semantic segmentation of brain tumors in mri images. In International MICCAI Brainlesion Workshop, pages 272–284. Springer, 2022.

