# OpenReview forum: "Multi-modal Cell Segmentation based on U-Net++ and Attention Gate"
_NeurIPS.cc/2022/Challenge/CellSeg — Submitted to NeurIPS CellSeg 2022_

### Official Review · Reviewer_DAFc · 2022-12-26
**This article does not show enough contributions and I do not recommend accepting it.**

**Rating:** 5
**Confidence:** 4

**Review:**

## Summary

The authors proposed a model that combines the attention gates with U-Net++ for cell instance segmentation with various modalities. The network can improve the segmentation performance compared to some other networks. They also show that quantity of attention gate and arithmetic of post-processing in their pipeline is important for the ultimate performance.

## Pros

The experiments are well organized and analyzed in detail.

## Cons

1. It is better to condiser utilizing unlabeled images in the pipeline.
2. UNet, U-NetR, Swin U-NetR and U-Net++ cannot represent SOTA of cell segmentation algorithms.
3. Their pipeline does not achieve good performance, more strategies can be explored such as pretraining with public datasets.

---

> ### Author Response · Authors · 2023-01-27
> **Some questions about your proposal**
>
> Thank you for your suggestions！I have a question about your suggestions. what does the “utilizing unlabeled images in the pipeline” mean exactly?

---

> > ### Comment · Reviewer_DAFc · 2023-01-30
> > **Answer**
> >
> > In your paper, only the labeled data are used for training, which may limit the performance and generalizability of your pipeline. You can read some paper about semi-supervised learning and integrate these strategies into your pipeline.

---

### Official Review · Reviewer_CfwZ · 2022-12-27
**The paper lacks of contribution and essential results.**

**Rating:** 4
**Confidence:** 4

**Review:**

Summary:
This paper attempts to improve the U-Net++ segmentation model by combining it with attention gates. The ablation study verifies the effectiveness of the modification.

Pros:
+ The figure of the network architecture looks good.

Cons:
+ The paper's content is simple and misses some critical parts, e.,g. efficiency results.
+ The method is naive and performs relatively inferior to other cell segmentation methods.
+ The motivation for adding attention gates to U-Net++  needs to be clarified.

---

### Official Review · Reviewer_x6Yg · 2023-01-07
**The paper has not well investigated the state-of-the-art and the improvement is not robust**

**Rating:** 2
**Confidence:** 5

**Review:**

Summary:
This paper proposed a network that combines attention gates with U-Net++ to segment multi-modal cell images. The method is clearly illustrated but the improvement is not robust. And the performance of this method is far from the state-of-the-art. In my opinion, this paper should be rejected.

Pros:
1. The authors use attention gates to improve a UNet++ model by highlighting the important area of feature maps.

Cons:
1. The authors over-claim that their method is superior to state-of-the-art networks. However, the baseline methods in the experiment are not the state-of-the-art and the performance of their proposed method is also far from the state-of-the-art.
2. The authors failed to investigate the latest competitive cell/nucleus segmentation methods. Thus, they also failed to reproduce and compare with the most competitive methods.
3. Attention gate seems to an important part in the paper but its implementation details are missing. It is suggested to present attention gates via formulas.
4. In Fig.4, why place the attention gates at these three positions? Experimental results on the setting are missing.
5. The best result in Tab.1 is 0.5637 and the best one in Tab.2 is 0.5972. Why they are different? Are Tab.1 and Tab.2 using different testing data? These experimental details are important but they are missing. I guess Tab.1 is obtained on the official website and Tab.2 is obtained on the validation set, right?
6. The improvement of attention gates is of poor generalization. In Tab. 2, the attention gates can bring an improvement of about 3-4% (0.597 – 0.559). However, in Tab. 1, the improvement of attention gates is about 0.1%, by comparing U-Net++ (0.5622) with Proposed method (0.5637).
7. The details of post-processing are missing.

---

> ### Author Response · Authors · 2023-01-26
> **Some explanations about my paper**
>
> Thank you for your suggestions！I want to explain something about my paper.
> 1.The time I got the score 0.5972 is after the deadline for submitting the docker. So I couldn't update my best model docker.This is why my docker performance is inconsistent with the best performance of the table 2.
> 2.Table 1 and 2 used the same data, the difference result between the same net was caused by different training parameters.And one is to test between different networks, another is a ablation experiment, the best performance obtained was certainly different.
> 3.F1 scores were calculated on the Grand Challenge, I don't have any verification datasets.

---

### Official Review · Reviewer_8geB · 2023-01-13
**Nice figures but not enough description of contributions**

**Rating:** 5
**Confidence:** 3

**Review:**

### Summary
To solve the cell segmentation from diverse morphologies and impurities, the author combined attention gates with U-Net++ to segment varied sizes of cells. They improved the segmentation by using the attention gate's feature filtering, resulting in a 0.5874 F1 score.

### Strength
1. The experiments indicated that adding attention gates into U-Net++ is effective.
2. All figures were nicely drawn, and they explained the architecture clearly.

### Weakness
1. The author could add some qualitative results for attention gates, specifically to emphasize the motivation of U-Net++ with these gates.
2. The experiments need more detailed quantitative results on tuning/validation/test sets separately.
3. Did all parameters in Table 2 go through all parameters? Please tell us more details.

### Minor Comment
1. The author could tell more about the limitation of the proposed method.

---

> ### Author Response · Authors · 2023-01-27
> **Some questions about your suggestions**
>
> Thank you for your suggestions！I have some questions about them.
> 1.I don't know “qualitative results”exactly means. Can you give me some specific examples？
> 2.About table 2, it's just a ablation experiment.  The only parameter(or variable) is the number and position of attention gates. If you mean to go through all the situations, I explained at the end of the article that this would be a hard work

---

> > ### Comment · Reviewer_8geB · 2023-01-28
> > **Some possible examples**
> >
> > I have referred to the paper checklist https://www.overleaf.com/read/xbnttbtmdkht. Please read that for more information.
> > 1. As I wrote above, the qualitative results for the almost same U-Net w/ and w/o attention gates. For example, show at least two examples with good segmentation results and two with poor segmentation examples. It is better to explain why these attention gates will improve the specific position.
> > 2. For reproducibility, please write more about how to search for the number and position of attention gates. For example, how to determine the starts, ends and steps of the grid search (if you use this kind of strategy).

---

### Decision · Program_Chairs · 2023-01-19

Accept